# Optimizing health service location in a highly urbanized city: Multi criteria decision making and P-Median problem models for public hospitals in Jeddah City, KSA

Abdulkader Murad[1]*, Fazlay Faruque[2], Ammar Naji[1], Alok Tiwari[1], Emad Qurnfulah[1], Mahfuzur Rahman[3], Ashraf Dewan[4]

1 Department of Urban and Regional Planning, Faculty of Architecture and Planning, King Abdulaziz University, Jeddah, KSA, 2 Department of Preventive Medicine, UMMC, Jackson, Mississippi, United States of America, 3 Department of Civil Engineering, International University of Business Agriculture and Technology, Dhaka, Bangladesh, 4 Spatial Sciences Discipline, School of Earth and Planetary Sciences, Curtin University, Bentley, Western Australia, Australia

* amurad@kau.edu.sa

**Data Availability Statement:** This research article does not present new data. The study is based on a comprehensive literature review and analysis of

## Abstract

Rapid urbanization and population growth have increased the need for optimizing the location of health services in highly urbanized countries like Kingdom of Saudi Arabia (KSA). This study employs a multiple-criteria decision making (MCDM) approach, e.g., fuzzy overlay technique by combining the P-Median location-allocation model, for optimizing health services. First, a geodatabase, containing public hospitals, road networks and population districts, was prepared. Next, we investigated the location and services of five public hospitals in Jeddah city of KSA, by using a MCDM model that included a fuzzy overlay technique with a location-allocation model. The results showed that the allocated five hospitals served 94 out of 110 districts in the study area. Our results suggested additional hospitals must be added to ensure that the entire city is covered with timely hospital services. To improve the existing situation, we prioritized demand locations using the maximize coverage (MC) location problem model. We then used the P-Median function to find the optimal locations of hospitals, and then combined these two methods to create the MC-P-Median optimizer. This optimizer eliminated any unallocated or redundant information. Health planners can use this model to determine the best locations for public hospitals in Jeddah city and similar settings.

## 1. Introduction

Important factors influencing the direct and societal expenses of healthcare services in an area include the optimal location of healthcare facilities and the distribution of patients [1]. These broad strategic choices also affect quick decisions like allocating resources and setting priorities. Choosing a site for a facility is a long-term investment, and once committed, it may be costly to relocate or alter the location. In addition, population growth and urbanization may make current policies less ideal in future. Hence, a city's incredible expansion propelled by

existing data sources. No original data were generated or collected for this research.

**Funding:** This project was funded by the Deanship of Scientific Research (DSR), King Abdulaziz University, Jeddah, Saudi Arabia, under grant no. (KEP-8-137-41). The authors, therefore, acknowledge with thanks DSR for their technical and financial support. The funders had no role in study design, data collection and analysis, decision to publish, or preparation of the manuscript.

**Competing interests:** The authors have declared that no competing interests exist.

rapid urbanization and population growth has increased the need for optimizing health facilities.

The need for reasonable and accessible healthcare facility design has received increased interest in recent time [2]. Providing high-quality healthcare in emerging cities and surrounding regions is vital to understanding the healthcare system's growth process. Locating healthcare centers in inappropriate locations would be inefficient in providing better service to people, especially during emergencies. Therefore, appropriate locations promote patient accessibility and enhance the quality of services. Several studies have confirmed the pertinence of location in healthcare services, especially in minimising response time [3]. However, choosing the most optimal location of facilities and allocating clients to those are crucial steps in designing a healthcare distribution network. Location-allocation models (LAMs) provide a suitable choice for this kind of decision-making challenge. LAMs include population or demand locations, facility locations, and travel distance or time between facilities and demands.

Numerous studies have examined the challenges of allocating resources to meet changeable demand resulting from the population boom and relocating facilities, e.g., warehouse location, saving client's travel time [4]; improving location of public schools [5]; placement of food pantries [6]; optimization of periodic distribution network [7]; placing fire stations [8]; emergency evacuation centers for natural hazard victims [9, 10]; and optimized location-allocation of earthquake relief centers [11]. In addition, many studies have also examined healthcare locational issues [12–16].

Evidence-based planning of hospitals, medical centers, and other health facilities using state-of-the-art technique is becoming invaluable to warrant the provision of sufficiently available, cost-effective, and high-quality healthcare in rural and urban regions across the world. The complete results, well-grounded suggestions, and recommendations of such investigations offer promising outcomes to health managers and policymakers in both developed and developing countries. Geographic information systems (GIS) and other spatial methods in health planning answer fundamental concerns regarding spatial analytics and incorporate multiple geographic perspectives of individuals, health care infrastructures and the environment. A GIS has become a useful tool with specific advantages as it can facilitate complex analyses [14]. Improvements in healthcare planning may be made by examining early reactions of patient flows using distance optimization simulations and healthcare scenarios in a geographical context. This optimization method integrates demographic characteristics, patient records, hospital information, and transportation networks using normative location/allocation modelling [15].

Policymakers need to consider various socioeconomic [16] and infrastructural factors [17], including distance to healthcare facilities, the distribution of population, availability and number of beds, size of hospitals, number of patients, and median household income to construct new hospitals. A GIS is an important tool in determining the optimum location of facilities [18]. The current work optimizes the location of hospitals by combining MCDM-based fuzzy overlay and P-Median LAM tools with maximize coverage (MC) location problem. The LAM is an optimization model used to determine the best location to allocate resources, such as facilities or services, to meet the demands of a particular population. This model aims to minimize the cost of providing these resources while ensuring they are adequately distributed across space to satisfy the needs of the populations. The model has two components; location and allocation. The location component involves determining the optimal location for a facility or service. The allocation part determines how to distribute resources to the population to meet their needs based on the proximity to the facility or service. The LAM model can be used in various fields, including healthcare, transportation, and emergency services. For example, it can be used in healthcare to determine the optimal location of hospitals, clinics, or pharmacies

to serve a particular population. The model can also be used to allocate medical resources, such as doctors, nurses, and medical equipment, to different locations based on the needs of the populations.

This work has the following research questions: (i) How can healthcare centers be located and fulfil the demand during an emergency? and (ii) how can demand be ensured equitable facilities? While models of geographic accessibility help evaluate existing public healthcare delivery systems, they do not guide where resources should be allocated or relocated. This issue may be overcome by integrating LAMs with MCDM in a GIS. Besides, healthcare planning can be better planned using P-Median and MC models [19–22].

This study aims to demonstrate capabilities of spatial analytics to compute optimum hospital service areas using LAM and fuzzy overlay technique in Jeddah City, KSA. It sheds light on how healthcare facilities should be best located in densely populated places like Jeddah. This study aims to contribute to the fields of urban planning and healthcare management. It helps stakeholders, including researchers and health professionals, better use limited healthcare resources to meet the needs of a growing population. The study also provides evidence for the utility of MCDM methods in the healthcare sector. It demonstrates maximizing healthcare locations by integrating geospatial data with population demographics, road networks, and other factors. This information may pave the way for further studies and improved methods of urban planning and support resource allocation.

Rapid urbanization and population increase may have far-reaching social consequences, including the exhaustion of healthcare infrastructure and the inability of certain populations to get the treatment they need. Recommendations from this work might greatly enhance healthcare access and equality for the people of Jeddah City by finding the most appropriate places for constructing future hospitals. Besides, this study has wider ramifications than only for Jeddah City. The methods and insights presented in this work can be a useful reference for health planners and policymakers beyond the study area. According to the literature, a study in São Paulo [1] concluded that access to healthcare is important for people, especially those in low-income groups. The study identified various barriers to healthcare accessibility, including proximity, safety, and quality of care. They recommended that planners need to integrate transport and health policies to tackle health inequalities. Another relevant study in Shenzhen City, China [2] found that an appropriate adjustment of general hospital location could significantly improve healthcare equity. Likewise, a study from Irbid municipality, Jordan [3] confirmed that the optimum location of healthcare services would improve the quality of services, including spatial accessibility and the patient-doctor capacity ratio.

The novelty of this study is the use of spatial analytics to produce an optimized hospital facilities area in Jeddah City. Specifically, the use of LAM and fuzzy analytical techniques is a unique approach that has not been extensively studied in the context of healthcare planning in Saudi Arabia or elsewhere. Furthermore, fuzzy overlay allows assessing the viability of potential locations and identifying ideal spots for healthcare facilities. In addition, it allows a multicriteria analysis that considers the possibility of a phenomenon belonging to multiple sets, which is useful when a location must meet specific criteria to be suitable. Additionally, the approach considers both reduced costs and optimized travel time as goals of LAM, to ensure that healthcare facilities are efficiently located to serve high-demand locations within a specified threshold followed by less-demand sites. Overall, the approach of this study can benefit relevant stakeholders, such as healthcare providers and policymakers, by providing a more efficient and effective way to plan and allocate healthcare resources in Jeddah and beyond.

## 2. Materials and methods

### 2.1 The study area

Jeddah, the second-largest city in KSA, is used as a case because firstly, it is experiencing a higher population growth compared to other cities in KSA (with its population doubling over the past 20 years). In 2020, Jeddah's population was over 3.4 million, which was only 2.3 million in the year 2020. During 2000 to 2010 decennial population growth rate was as high as 56.4%, indicating a significant increase in healthcare services demand, and making it an excellent location to study hospital placement [23, 24]. Additionally, Jeddah is a major economic hub and serves as a gateway to the holy cities of Makkah and Madinah, attracting many tourists, business travellers, and expatriates who require access to quality healthcare services. Economic significance of Jeddah [25] also makes it a useful location for studying hospital placement.

Jeddah is a regional healthcare center, with several major hospitals and medical centers (Fig 1). Studying hospital placement in Jeddah can provide valuable insights about overall healthcare system in KSA and the region, as well as the challenges and opportunities associated with hospital placement in a regional medical center. Moreover, Jeddah is prone to natural disasters such as flooding [26], which can significantly impact healthcare services. Thus, studying hospital placement in Jeddah can help identify optimal locations that can withstand natural disasters and ensure continuous healthcare services. Finally, Jeddah's diverse population includes a large expatriate community with varying healthcare needs, providing an opportunity to study how hospital placement can address the unique healthcare needs of different communities.

### 2.2 Data

We used several spatial and non-spatial datasets to identify optimum hospital locations and respective service areas. The databases include features for public hospitals, road networks, and population districts, obtained from respective government agencies (Table 1). Each feature is associated with pertinent attributes that support developing the required models.

### 2.3 Data analysis with MCDM process

**2.3.1 Fuzzy overlay.** Fuzzy logic and fuzzy set theory, first introduced in 1965 by Zadeh (1965), are extensively used in uncertainty modeling and in decision-making [27]. In this study, fuzzy overlay is utilized to estimate the spatial distribution of hospitals in the city. This process includes several steps: (i) selecting target layers that reflect site characteristics, which is used in deciding best locations for hospitals, including the number of beds and population density; (ii) assigning fuzzy membership values, on a scale of 1 to 0, to each layer, based on relevant parameters; and (iii) combining fuzzy layers to determine logical operators.

In general, high-score places are the best and most fitting. Once the model is validated by hospital locations and the findings assessing its accuracy and the need for modification are reviewed, suitable locations can be identified, and the results can be applied to view the final suitability layer by fuzzy membership functions.

The fuzzy overlay tool also allows analysis of the possibility of a phenomenon belonging to multiple sets in a multi-criteria analysis. The fuzzy analysis covers the following:

*2.3.1.a Spatial distribution of hospitals.* Geographical viewing starts with points of interest and asks about attributes of events located within area of interest. Fig 1 shows the location of hospitals, describing the distribution of five public hospitals in the city. The sizes of these hospitals are not the same. For example, King Fahad Hospital, located in the central part, has a

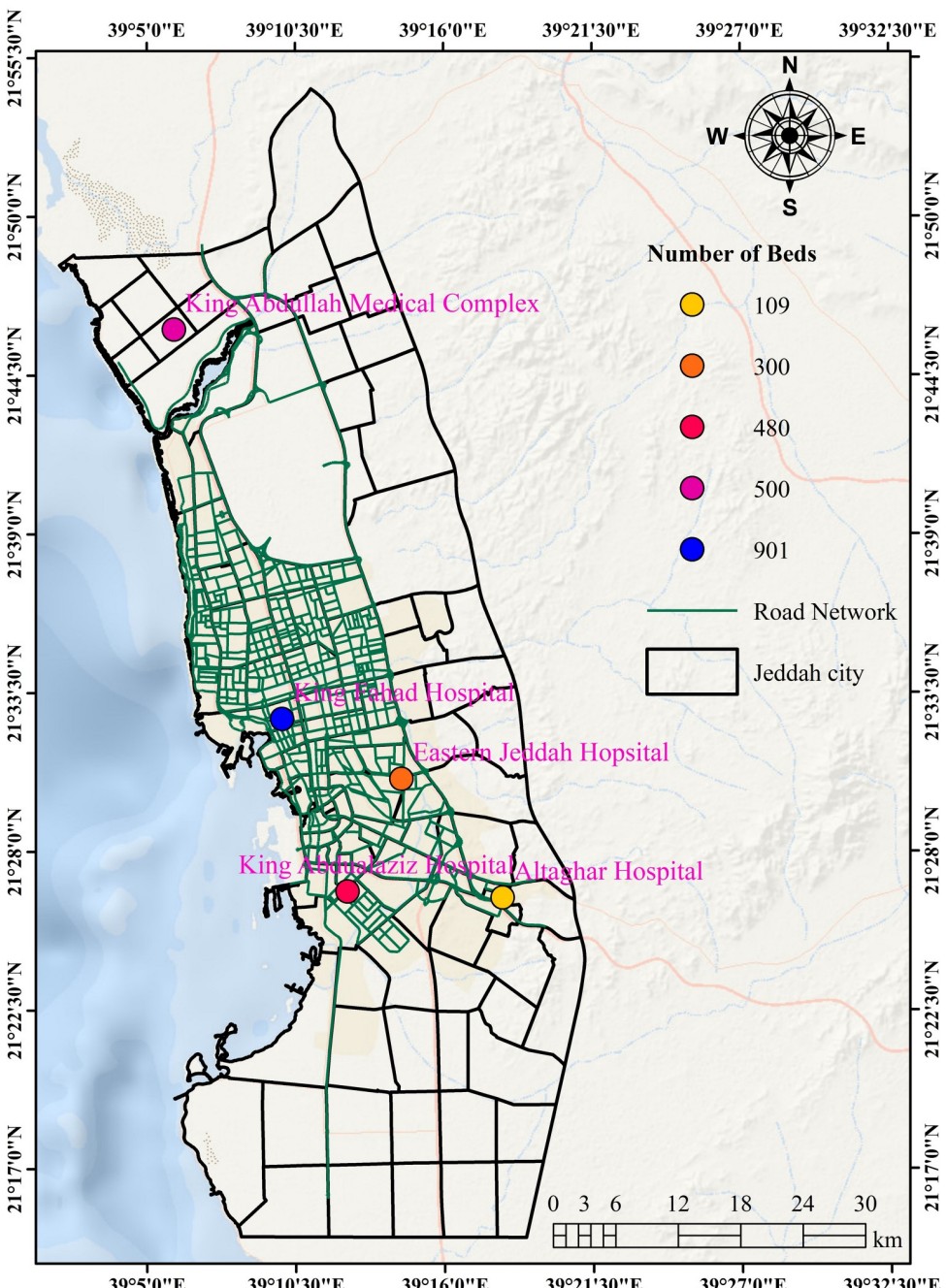

**Fig 1. Location of public hospitals with their bed capacity in Jeddah City, KSA (The figure is made with ArcGIS software (https://www.esri.com/)).** [Shapefile or base map is freely available at https://www.diva-gis.org/gdata and does not have copyright restrictions].

capacity of about 900 beds. Meanwhile, Al-Thaqar Hospital, located in the south of the city, contains as low as 109 beds.

*2.3.1.b Kernel density estimation.* Kernel density can be estimated from either points or line features using a non-parametric approach. This method is extensively used in various

**Table 1. Data used in this study.**

| Data | Detail | Feature type | Source | Process |
|------|--------|-------------|--------|---------|
| Public hospitals | Location | Point | Ministry of Health, Jeddah (2020) | Determining location of facilities |
| Road networks | Connectivity | Linear | Jeddah municipality (2018) | Explaining service network |
| Districts | District boundaries | Polygon | Jeddah municipality (2018) | Deriving extent of service area |
| Hospital beds | Availability of beds | Numeric | Ministry of Health, Jeddah (2020) | Finding density of health services and classifying public hospitals |
| Population | Number of people in each district | Numeric | Jeddah municipality (2018) | Obtaining set of demand points of health users |

applications, including risk mapping and hotspot identification. Kernel functions are derived from quadratic model explained by Silverman (2018) [28]. We use point features as an input to the kernel density, which includes spatial filtering by a search window (also called bandwidth). For point data, a kernel can give each point a "spatial meaning" and estimate the variable's probability distribution at each location within a study area. The output dimensions for the associated cells contribute to the definition of the kernel as a three-dimensional feature.

*2.3.1.c Location-allocation modelling.* In this study, the LAM toolset is used to determine optimal hospital locations for the surrounding populations. We used the P-Median tool as it selects optimal locations based on impedance to the facility (e.g., hospital, Fig 2). A modified P-Median model, which considers both geographic accessibility and service quality, is proposed that employs both exact and approximate strategies [29]. Another example is found in the application of this model in a case study in Hainan Province, China, resulting in the selection of three optimal healthcare centers from candidate cities and allocating resources considering capacity constraints and spatial compactness constraints [30].

Facilities such as hospitals can reduce costs and keep their accessibility high using LAM. The following six issues can be addressed through LAM analysis: minimize impedance, maximize coverage, minimize facilities, maximize attendance, maximize market share, and target market share. The P-Median problem model produces service catchment area and allocates demand locations. The P-Median clustering model has its roots in operations research, originating from efforts to optimize the planning of facility locations.

## 3. Results and discussion

Fig 3a, 3b shows the results of kernel density analysis for hospital beds and population. The former identifies bed density and reveals that the density is higher in the downtown area but relatively low in the north and east of the city. The latter demonstrates population density, which increases in the downtown area and to the north and east of the city. Meanwhile, the southern and eastern city districts are having low population densities.

The present study has selected the fuzzy 'And' model to get the maximum value from all inputs. In this case, all inputs must have a high value to obtain a high output value. This means that cells located near higher population density and, at the same time, having higher bed density will indicate locations of high provider-to-population ratio with better hospital services than locations with lower provider-to-population ratio.

Fig 3c defines these less-served locations in the city, which are mainly in the south, east, and north. Health planners should prioritize these locations to increase optimized hospital locations in Jeddah City. Health planners can use this output to help decide where to build a new hospital. This work suggests local health planners to assign new hospitals in these locations to get better health services in the city.

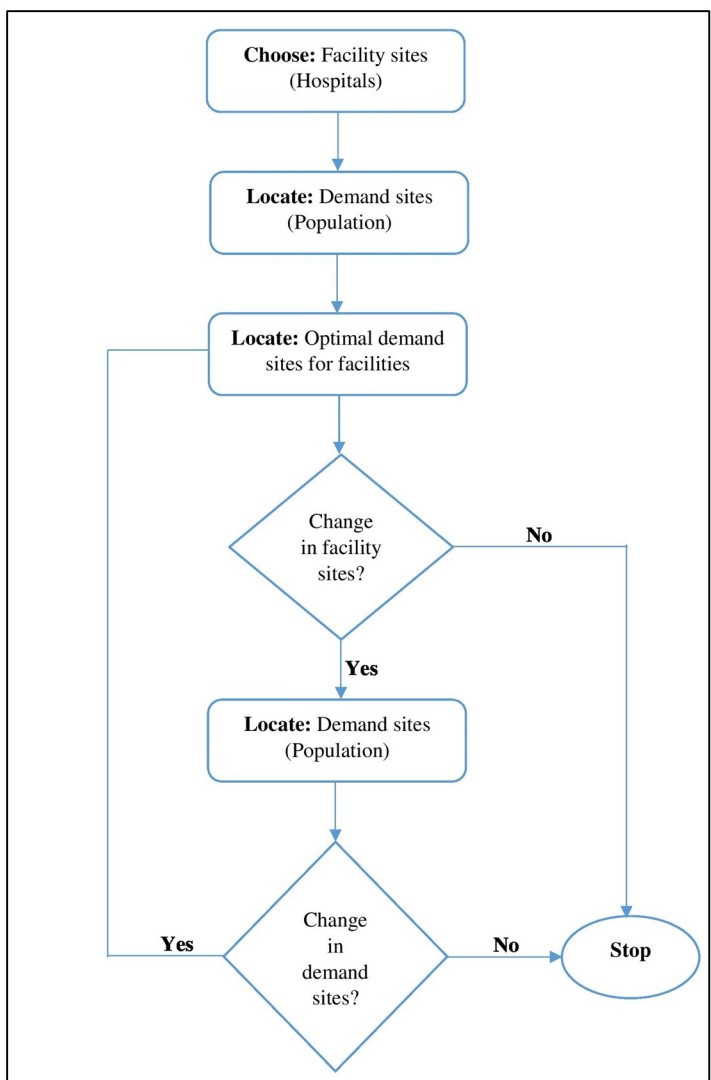

**Fig 2. Schematic diagram, showing P-Median solution (The figure is made with Microsoft word v.2016 (https://www.microsoft.com/en-us/download)).**

The P-Median model was set up to choose the best facilities that could serve the most-demand districts within 6-hour of the nearest hospitals (facilities). We have assumed that all facilities are equally important, so each receives a weight of 1. The choice of the 6-hour impedance cut-off in the LAM may have been based on several factors, including practical considerations, such as available resources, population density, and the nature of the demand locations. For example, a 6-hour impedance cut-off may have been selected because it is a reasonable travel time for patients seeking medical services and covers a significant portion of the study area. It may have been chosen because it is a manageable distance for healthcare providers to travel to reach the target locations. In addition, the 6-hour impedance cut-off may have been chosen because it reveals peak demand locations, which may be critical in identifying areas that require immediate attention or additional resources. However, other impedance cut-offs could also have been considered, such as 3 hours, 4 hours, or 5 hours. The choice of the

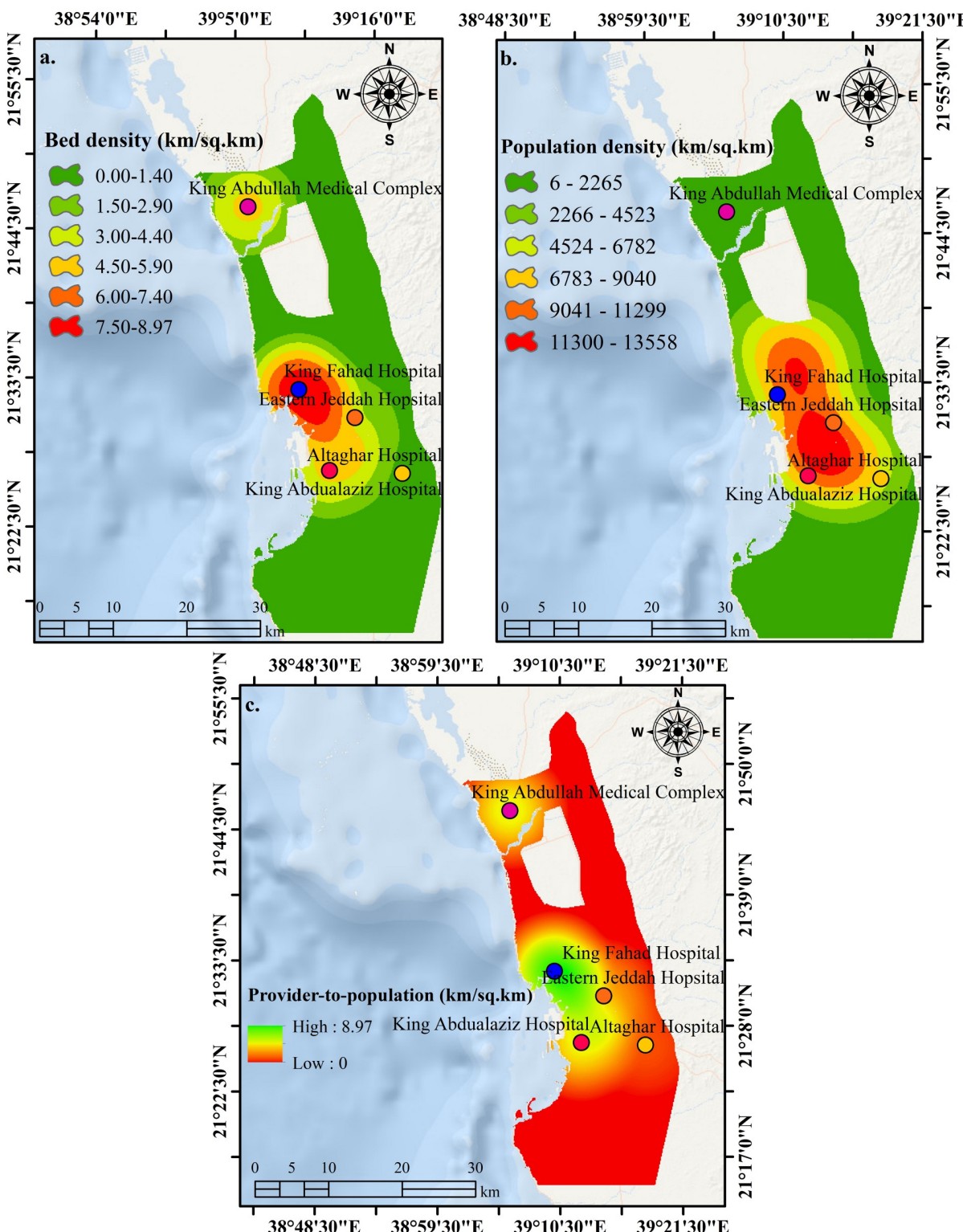

**Fig 3. a.** Number of beds in hospital derived by kernel density function. **b.** Population density in Jeddah as a function of health demand. **c.** A provider-to-population result-based fuzzy overlay analysis. The public hospitals better serve locations in red, while locations in green need additional hospitals (These figures are made with ArcGIS software (https://www.esri.com/)). [Shape file or base map is freely available at https://www.diva-gis.org/gdata and does not have copyright restrictions].

impedance cut-off ultimately depends on specific context, objectives, and constraints of the location-allocation problem. The results of the P-Median model reveal that 94 districts out of 110 are covered by the hospitals, where the King Abdualaziz Hospital served 22 districts. The analysis shows the optimum catchment area for this hospital, which helps local health planners to focus on these 22 districts in terms of providing the required healthcare services for all residents living in these districts.

According to this study, demand locations should be covered within the specified coverage time of 6-hour from the nearest facilities, and highly neediest districts should be served within an hour's drive from the hospitals. Coverage optimization of facilities utilizing standalone LAM can be complicated due to unpredictability of demand locations routing problems for services. Therefore, we applied an optimized MC-P-Median problem model to address these shortcomings. We use several impedance factors to evaluate coverage of demand locations within impedance thresholds (cf. optimized maximize coverage location problem model to select facility locations).

After applying P-Median LAM, it was determined that five hospitals (facilities) were unable to reach all districts (110 districts) within the specified travel time due to unallocated and redundant information (Fig 4). Therefore, the P-Median problem model was optimized with the MC location problem model. The MC location problem model attempts to optimize the distribution of the potential locations for an assigned travel time. The MC model solves problems that serve the highest demand location, based on distance or time. Besides, the MC model does not minimize the number of hospitals (facility locations) needed to cover all districts over specific distances or times. It helps to solve the redundant information of the LA problem. The results of the P-Median problem model indicate that there were 16 demand locations inaccessible or redundant (Fig 4). In the optimization process, redundant demand locations were eliminated to cover the maximum demand locations within a short time, owing to the fact that it can be challenging to optimize the demand locations by employing standalone P-Median model because of associated uncertainty (Fig 5). The results reveal that ~45% of demand locations could be accessed within a travel time of 1.5 hours to cover districts of the city (Table 2). This proportion rises as the impedance cut-off rises to 6 hours, i.e., the impedance cut-off reveals peak demand locations (94 out of 94 demand locations, Table 2 and Fig 5). This outcome can be used to visualize the aim of this work, which can further be optimized by considering the capacity of each hospital. If the capacity of a hospital was known, this optimized model could identify if further facilities were needed for the study area.

In the last few decades, GIS has increasingly been used as a critical spatial decision support system (SDSS) for evaluating suitable locations of healthcare services. Although several methods are available within the geospatial community to identify suitable locations of hospitals, we used fuzzy overlay and LAM techniques in this work.

As the P-Median LA model alone cannot handle uncertainty related to real numbers, we used a fuzzy overlay to reduce uncertainty. According to the P-Median model results, existing five hospitals were insufficient to address all demand points within the specified travel time of 6 hours. Particularly, 16 districts were inaccessible due to unlocated issue and redundant information. To choose the best distribution of hospitals across Jeddah City and to ensure better coverage of demand location, an optimized model considering MC and P-Median models was applied. Such a combination, including MC-P-Median and fuzzy overlay, is rarely used in the health planning sector. The models applied in this study are expected to contribute to overcoming the problem of exponential increase in patients' accessibility to hospitals in Jeddah City. These models significantly improved the results of hospital accessibility over traditional overlay index methods, such as buffer analysis or distance-based catchment area delineation.

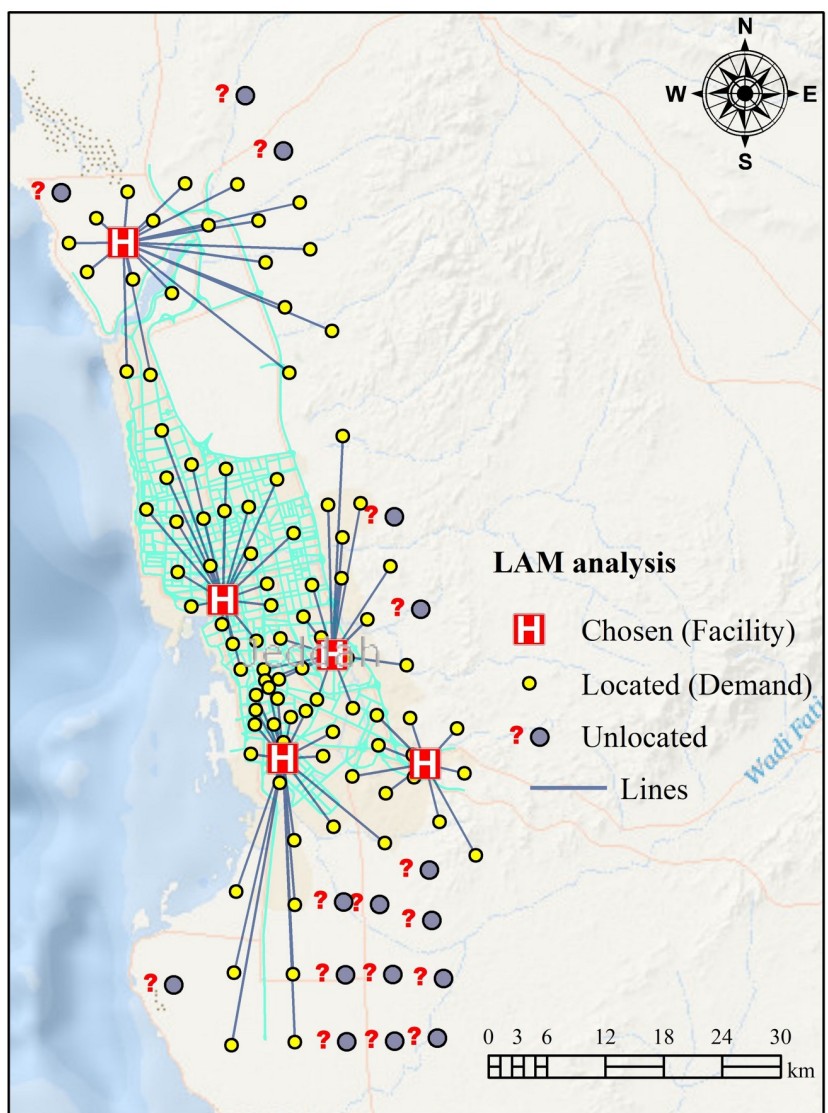

**Fig 4. LAM for hospitals, based on the P-Median problem (here, straight lines connect each demand-point/facility pair).** This figure is made using ArcGIS software (https://www.esri.com/). [Shape file or base map is freely available at https://www.diva-gis.org/gdata and does not have copyright restrictions].

The solution to the optimized model reveals that expanding the impedance threshold enables us to cover a more significant number of districts. According to our findings, raising the impedance threshold can improve the coverage of unconnected demand nodes. The analysis presented here may help satisfy the interest of the decision-makers in moving some of the centers to redistribute the healthcare facilities.

The study has wider ramifications given the global surge in urban populations. Although the results of a LA model are often limited to the study area and may not be readily transferable to other areas or cities, we believe the method can be scaled up to other settings. Besides, the study appears to be limited to the boundaries of Jeddah City, despite the existence of other adjacent localities in the problem set. By excluding adjacent localities from the study area, the results may not fully capture the broader healthcare needs and dynamics of the entire region.

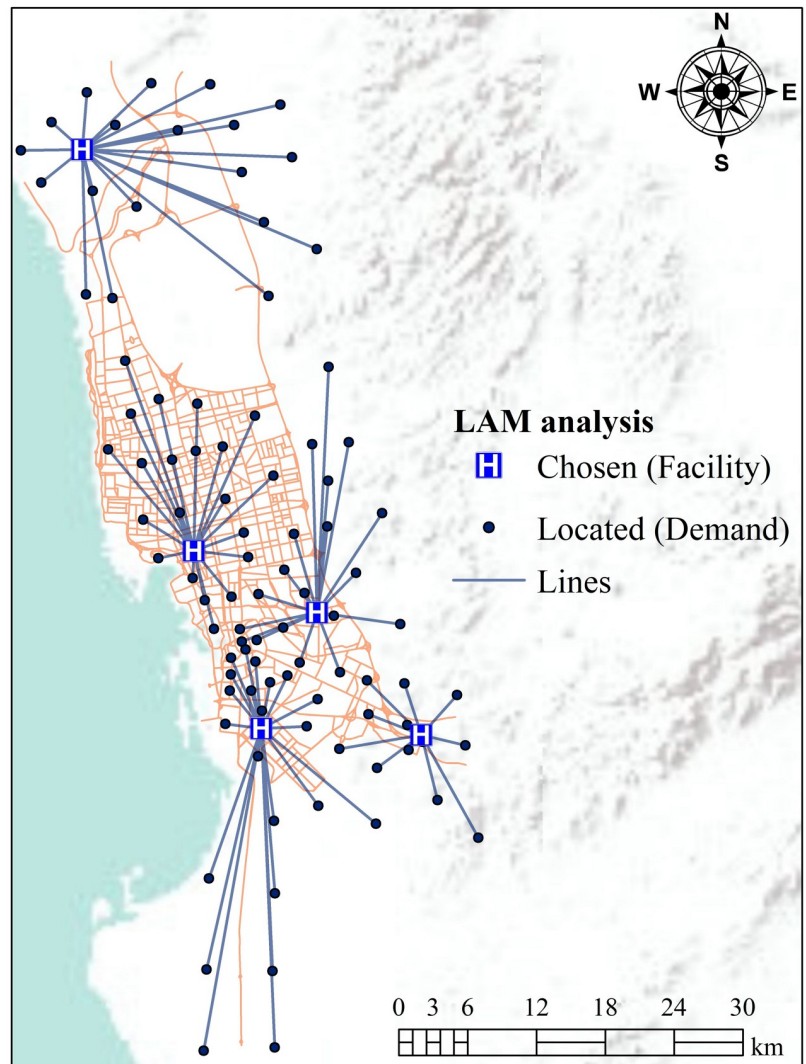

**Fig 5. LAM for hospitals, based on optimized P-Median considering MC location problem model (here straight lines connect each demand-point/facility pair).** This figure is made with ArcGIS software (https://www.esri.com/en-us/). [Shape file or base map is freely available at https://www.diva-gis.org/gdata and does not have copyright restrictions].

**Table 2. MC-P Median-based optimization considering different impedance cut-offs (shaded cells indicate maximum demand locations for each facility).**

| Facility (Hospitals) | Demand (Districts) | Percentage (%) | Demand and inaccessible locations with impedance cut-off (hour) | | | | | | | | | | | |
|---|---|---|---|---|---|---|---|---|---|---|---|---|---|---|
| | | | 0.5 | 1.0 | 1.5 | 2.0 | 2.5 | 3.0 | 3.5 | 4.0 | 4.5 | 5.0 | 5.5 | 6.0 |
| H1 | 21 | 22.34 | 0 | 3 | 9 | 11 | 15 | 16 | 20 | 21 | 21 | 21 | 21 | 21 |
| H2 | 19 | 20.21 | 1 | 3 | 10 | 15 | 16 | 18 | 18 | 19 | 19 | 19 | 19 | 19 |
| H3 | 22 | 23.40 | 1 | 6 | 11 | 16 | 16 | 18 | 18 | 18 | 20 | 20 | 20 | 22 |
| H4 | 20 | 21.28 | 2 | 2 | 2 | 2 | 3 | 8 | 13 | 13 | 16 | 19 | 20 | 20 |
| H5 | 12 | 12.77 | 2 | 6 | 10 | 12 | 12 | 12 | 12 | 12 | 12 | 12 | 12 | 12 |
| **Total** | 94 | **Percentage (%)** | 6.38 | 21.28 | 44.68 | 59.57 | 65.96 | 76.60 | 86.17 | 88.30 | 93.62 | 96.81 | 97.87 | 100.0 |

H1: King Fahad Hospital, H2: Eastern Jeddah Hospital, H3: King Abdualaziz Hospital, H4: King Abdullah Medical Complex, and H5: Altaghar Hospital

However, this limitation should not significantly affect the overall accuracy and reliability of the optimized locations for healthcare facilities, as the model may not account for potential demand from neighbouring areas or the influence of population movement across boundaries. However, to address this issue, further research is warranted.

We can infer that the solutions presented in the study are feasible as this model-based allocation of hospitals can serve substantially more districts of the study area. While the approach implemented in this study can improve healthcare services through better allocation, further optimization will be required to adjust to population dynamics by redistributing healthcare facilities and creating new facilities.

One of the key takeaways from this study is the knowledge gained through the integration of the P-Median model and fuzzy overlay technique, contributing to enhanced healthcare access via resource allocation. The results underscore the importance of considering factors, such as population density, proximity of healthcare facilities, and travel times in healthcare planning. This approach acknowledges that healthcare accessibility is not solely contingent on the number of hospitals but also on their strategic placement relative to population centers. Furthermore, the study highlights the significance of considering the local context, such as available resources, population density, and the nature of demand locations when determining travel time thresholds. Thus, adaptability and context-based decision-making are vital for effective healthcare planning.

While the results suggest the usefulness of our approach, we duly acknowledge the limitations of this study. This study may have generalizability problems, especially if it does not give insights into the transferability of the suggested paradigm to other situations. The study does not appropriately address the possible effect of infrastructure, dynamic changes in population, or socio-economic disparities that they may have on the findings. This limitation may affect comprehensive understanding of healthcare needs and dynamics in the broader region. The discussion should have the implications for not accounting for potential demand from neighbouring areas and the importance of regional dynamics in healthcare planning.

## 4. Conclusions

The results of the study in Jeddah City, using a combined LAM and fuzzy overlay method, demonstrated that about 45% of demand locations could be reached within a travel time of 1.5 hours. This proportion increases as the impedance cut-off (travel time threshold) raises to 6 hours. In fact, at this impedance cut-off, all demand locations could be reached (94 out of 94 demand locations). These results may help in enhancing the accessibility of each population site based on the improvement of nearby facilities and transportation modes.

In conclusion, this study offers a comprehensive approach to allocating and optimizing healthcare facility in urban areas. The findings underscore the importance of data-driven and context-specific strategies in addressing complex issue associated with healthcare accessibility for growing urban populations. Future studies might look at how multi-objective location-allocation issues can be used to address the above-mentioned problem.

## Supporting information

**S1 Data.**
(ZIP)

## Author Contributions

**Conceptualization:** Ammar Naji, Alok Tiwari, Mahfuzur Rahman.

**Data curation:** Abdulkader Murad, Mahfuzur Rahman.

**Formal analysis:** Abdulkader Murad, Mahfuzur Rahman.

**Funding acquisition:** Abdulkader Murad, Ammar Naji.

**Investigation:** Emad Qurnfulah.

**Methodology:** Abdulkader Murad, Fazlay Faruque, Alok Tiwari, Emad Qurnfulah, Mahfuzur Rahman, Ashraf Dewan.

**Project administration:** Abdulkader Murad, Ammar Naji.

**Resources:** Emad Qurnfulah.

**Software:** Mahfuzur Rahman, Ashraf Dewan.

**Supervision:** Emad Qurnfulah.

**Validation:** Ammar Naji, Alok Tiwari, Ashraf Dewan.

**Visualization:** Mahfuzur Rahman, Ashraf Dewan.

**Writing – original draft:** Abdulkader Murad, Fazlay Faruque, Alok Tiwari, Mahfuzur Rahman, Ashraf Dewan.

**Writing – review & editing:** Abdulkader Murad, Fazlay Faruque, Alok Tiwari, Mahfuzur Rahman, Ashraf Dewan.

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
