## [Decision Letter · Decision Letter 0]

5 Jun 2023

PONE-D-23-14949Journal name: Plos One

Optimizing Health Service Location in Highly Urbanized Countries:

Multi Criteria -P-median model for Public Hospitals in Jeddah City, Saudi ArabiaPLOS ONE

Dear Dr. Murad,

Thank you for submitting your manuscript to PLOS ONE. After careful consideration, we feel that it has merit but does not fully meet PLOS ONE’s publication criteria as it currently stands. Therefore, we invite you to submit a revised version of the manuscript that addresses the points raised during the review process.

We look forward to receiving your revised manuscript.

Kind regards,

Mohammed Sarfaraz Gani Adnan, PhD

Academic Editor

PLOS ONE

Journal Requirements:

"This paper is funded by the Deanship of Scientific Research (DSR), King Abdulaziz University, Jeddah, Saudi Arabia, under grant no. (KEP-8-137-41). The authors, therefore, acknowledge with thanks DSR for their technical and financial support."

"The authors declare no conflict of interest. "

6. We note that all Figures in your submission contain [map/satellite] images which may be copyrighted. All PLOS content is published under the Creative Commons Attribution License (CC BY 4.0), which means that the manuscript, images, and Supporting Information files will be freely available online, and any third party is permitted to access, download, copy, distribute, and use these materials in any way, even commercially, with proper attribution. For these reasons, we cannot publish previously copyrighted maps or satellite images created using proprietary data, such as Google software (Google Maps, Street View, and Earth). For more information, see our copyright guidelines: http://journals.plos.org/plosone/s/licenses-and-copyright.

a. You may seek permission from the original copyright holder of all Figures to publish the content specifically under the CC BY 4.0 license.  

Reviewers' comments:

Reviewer's Responses to Questions

**Comments to the Author**

1. Is the manuscript technically sound, and do the data support the conclusions?

Reviewer #1: Yes

Reviewer #2: Partly

Reviewer #3: Partly

2. Has the statistical analysis been performed appropriately and rigorously? 

Reviewer #1: Yes

Reviewer #2: Yes

Reviewer #3: I Don't Know

3. Have the authors made all data underlying the findings in their manuscript fully available?

Reviewer #1: Yes

Reviewer #2: No

Reviewer #3: No

4. Is the manuscript presented in an intelligible fashion and written in standard English?

Reviewer #1: Yes

Reviewer #2: Yes

Reviewer #3: No

5. Review Comments to the Author

Reviewer #1: My pleasure to review this work. The authors presented interesting findings of “Optimizing Health Service Location in Highly Urbanized Countries: Multi-Criteria -P-Median model for Public Hospitals in Jeddah City.” As climate change will bring changes in diseases geography everywhere, the topic is interesting and thus merit publication. However, there are some issues in the present work as it currently stands. I think considering those could further improve the quality of the manuscript before its publication. Overall, I would recommend a round of revision. My comments are outlined below.

1. The general description of the problem (Introduction) and the description of its importance for science and society may be further improved with some examples.

2. It will be better to add a study area map with lat. and long., so that international reader can get localtional information precisely.

3. Road features considered in this study are believed to be highly useful in developing a successful location-allocation model. Did you face any topological errors in the road network? If yes, then how you managed the error in the datasets? Please state it in your work.

4. Figures 1 and 2 unnecessarily increased the volume of the paper. I’d suggest combining them in a single figure.

5. Are the results (or the method) sensitive to this specific study area or can be scaled up?

6. Regarding the solutions, are these solutions Feasible or Optimal solutions? Please make this distinction both in the results and in their discussion.

7. Please make sure your conclusions section underscores the scientific value added to your paper and/or the applicability of your findings/results. It would help if you enhanced your contributions and limitations, underscored the scientific value added to your paper, and/or the applicability of your findings/results and future study in this session.

Reviewer #2: I would like to thank the authors for this interesting work. Please consider shortening the paper by avoiding repetition and too much theoretical descriptions that could be resolved using brief description and references to original studies introducing the theories/concepts. My detailed comments can be found below:

A)Abstract: “better serve better 94 districts” – please check

B)Introduction: The introduction is needlessly long. It would be better to remove some components that explain the applied model to the methodology section. The parts that could be considered for moving are listed below:

1. “This work proposed a novel method…facility or service”

2. “Based on the preceding discussion…facilities?” – the research question should come at the end of this section.

3. “The fuzzy overlay toolset…less-demand sites”

Additionally, please include a condensed literature review (a single para should suffice) that compares the existing LA model and how this method is novel or a new addition to the knowledge base. Moreover, the introduction does not have a meaningful flow. Please revise and reorganize.

C) Methods:

-Consider changing the heading of 2.1 to Study Area or a similar simpler heading

-2.1: First para requires some references

-2.1: A map of the study area could be useful

-2.3: Consider renaming the heading. Should be ‘Data Analysis’ or ‘MCDM Process’ or words similar to that effect. This is because all your sub-sections under section 2 are part of the methods.

-2.3: ArcGIS, which version?

-2.3: When explaining the algorithm that was used to determine the default kernel, the term ‘weighted’ was used. It is important to mention what this implies and what kind of weight assignments were performed.

-2.3.2: Condense. There are too many introductory descriptions here. Figure 1 could be placed in 2.1 to give general readers an idea about the study area before the variables are introduced in the subsequent parts. Jeddah’s boundary should be included in the Figure.

-2.3.2: Figure 1 and 2 could be merged. Consider showing the point locations of the hospitals and color gradation based on hospital numbers, as in Figure 2. The labels could be turned on to display hospital names on the points.

-2.3.4: Again, such a detailed description of the location-allocation model and its applications are unnecessary. Focus on the P-median model that you have applied, how it is novel, and how it was implemented in a GIS platform.

-2.3.4: Consider moving the mathematical description to a supplementary file. The paper is not method intensive. Therefore, too much focus on the mathematical model might distract the readers from the main objectives. Also, if you choose to include mathematical notations, there must be equation numbers, and every function (English alphabet) must be expressed with what they stand for. For example, it is unclear what the ‘a’, ‘d’, ‘x’ etc., stand for in the objective function. This description was provided at the end but every description must come underneath the equation when the function was first introduced. This part raises more questions than it answers. I would suggest simplifying this part and instead focusing on how this model was implemented by providing references to the P-median MCDM model from earlier studies as references.

-2.3.4: Instead of such a detailed mathematical description, consider including a schematic diagram and its description to explain the study design or P-median implementation.

- MCDM generally has a scaling factor applied to combine the multi-criteria variables. Please mention if this was done, and if yes, how.

-Figure 3: Scalebar has problems. The numbers should be spread out more in the beginning part.

3.2: First para repeats the methods. No need.

3.2: Third and fourth para could be used to replace section 2.3.4

Figure 6 (e.g., Candidate) and 7 (e.g., restriction) have symbols that do not exist in the map. Please revise. What does ‘lines’ mean?

Conclusion: Should be rewritten. The focus should not be on the application of GIS or fuzzy overlay or p-median. Rather, what you found in the study, specifically, what discrepancies in existing health facilities locations were revealed through the study analysis?

Reviewer #3: Title: Optimizing Health Service Location in Highly Urbanized Countries: Multi Criteria -P-median model for Public Hospitals in Jeddah City, Saudi Arabia Authors: Murad et al

Summary: The manuscript presents a novel approach for the optimization of hospital locations in urbanized regions like Jeddah City, Saudi Arabia, based on a multi-criteria decision model (MCDM) and the P-median location-allocation model. By combining these techniques into an MC-P-median optimizer, the authors aim to aid health planners in determining the best locations for public hospitals. The study was conducted using a geodatabase of public hospitals, road networks, and population districts in Jeddah, with results suggesting the need for additional hospitals for comprehensive coverage.

Major Concerns:

The major concern with this manuscript is its structure and organization, which presently do not facilitate a clear understanding of the work's aims, objectives, or hypotheses, particularly how these are addressed in the research findings.

Specifically, the authors should consider separating the results and discussion sections to clearly present and then interpret their findings. By doing so, they can more effectively contextualize their results within the broader literature, address the implications of their study, and discuss any limitations that might impact the applicability of their research in real-world settings.

Minor Observations:

• It's unclear from the manuscript whether the road network data was used in distance calculations.

• The figure captions need to be improved for better clarity and to guide the reader's focus within the figure.

• The authors mention localities adjacent to Jeddah City in the problem setting, but the study appears limited to the city's boundaries. The impact of this exclusion on the model's outcomes should be addressed.

• The text following the aim statement in the introduction seems redundant and could potentially be removed.

• The authors should consider merging Figures 1 and 2 to better convey the number of beds information. A consistent, rounded scale could also improve reader comprehension.

• On page 14, the formula for a linear programming problem should be contextualized for the current problem, explaining how various parameters in the equation are applicable in this context.

• The scale inconsistency across Figures 3 to 7 can be confusing for readers. The authors should ensure the scales remain consistent for the same aerial extent.

• Figures 3 to 5 lack internal reference points to aid reader comparison and understanding. The inclusion of recognizable features, such as significant points or road junctions, could improve this.

In conclusion, while the manuscript offers a promising contribution to the field of health service optimization, certain adjustments are necessary to enhance the clarity and impact of the findings. It is recommended that the authors carefully address the above concerns and observations in their revisions.

6. PLOS authors have the option to publish the peer review history of their article (what does this mean?). If published, this will include your full peer review and any attached files.

Reviewer #1: No

Reviewer #2: No

Reviewer #3: No

---

## [Author Response · Author response to Decision Letter 0]

23 Aug 2023

Response Letter

Dear Academic Editor, Mohammed Sarfaraz Gani Adnan, PhD

Thank you very much for allowing us to revise the manuscript. We greatly appreciate the editor’s and the reviewers constructive comments and suggestions on the manuscript entitled “Optimizing health service location in a highly urbanized country: multi criteria -P-Median model for public hospitals in Jeddah city, KSA.”

We have carefully reviewed the editor’s/reviewer’s comments and addressed almost them in the revised mansucript. All revised portions are marked in yellow in the revised manuscript, which we would like to submit for your kind consideration. The authors pay special thanks to the editor and reviewer because their comments and suggestions have greatly improved quality of the manuscript.

Regards

Prof Abdulkader Murad

Response to Reviewer 1 Comments

Comment: My pleasure to review this work. The authors presented interesting findings of “Optimizing Health Service Location in Highly Urbanized Countries: Multi-Criteria -P-Median model for Public Hospitals in Jeddah City.” As climate change will bring changes in diseases geography everywhere, the topic is interesting and thus merit publication. However, there are some issues in the present work as it currently stands. I think considering those could further improve the quality of the manuscript before its publication. Overall, I would recommend a round of revision. My comments are outlined below.

Response: The authors would like to thank the reviewer for suggestions. We are highly indebted to your comments, which improved quality of the manuscript. 

The manuscript has been rechecked and appropriate changes have been made in accordance with your suggestions. A response has been prepared to indicate changes in the revised manuscript relative to the comments. Corrected/inserted/amended texts are highlighted in YELLOW. 

Comment 1: The general description of the problem (Introduction) and the description of its importance for science and society may be further improved with some examples.

Response 1: Thank you for your valuable comments. We have revised the introduction section and described how science and society could be benefitted from this work (please see pages 5-6 and lines 100-126).

“Important for both science and society at large, this study sheds light on how healthcare facilities should be best located in densely populated places like Jeddah, KSA. This study makes a scholarly contribution to the fields of urban planning and healthcare management. It helps stakeholders, including researchers and health scientists, better use limited healthcare resources to meet the need of a growingpopulation. The study also provides evidence for the utility of MCDM methods in the healthcare sector. It demonstrates maximizing healthcare locations by integrating geospatial data with population demographics, road networks, and other factors. This information may pave the way for further studies and improved methods of urban planning and support resource allocation. Rapid urbanization and population increase may have far-reaching social consequences, including the exhaustion of healthcare infrastructure and the inability of certain populations to get the treatment they need. Recommendations from this research might greatly enhance healthcare access and equality for the people of Jeddah city by finding the most appropriate places for future hospitals. Besides, this study has wider ramifications than only for Jeddah city, KSA. The methods and insights presented in this research can be a useful reference for health planners and policymakers worldwide. In conclusion, this study can contribute to urban planning and healthcare management, and it is significant for society because of its potential to increase access to healthcare, improve public health outcomes, and inform policy decisions in similar urban settings. According to the literature, a study in São Paulo [1] concluded that access to healthcare is important for people, especially those in low-income groups. The study identified various barriers to healthcare accessibility, including proximity, safety, and quality of care. They recommended that planners need to design integrated transport and health policies to tackle health inequalities. Another relevant study in Shenzhen City, China [2], found that an appropriate adjustment of general hospital location could significantly improve healthcare services equity. Likewise, a study from Irbid municipality, Jordan [3], confirmed that optimum location of healthcare services would improve quality of services, including spatial accessibility and the patient-doctor capacity ratio.”

References

1. Guimarães, Thiago, Karen Lucas, and Paul Timms. "Understanding how low-income communities gain access to healthcare services: A qualitative study in São Paulo, Brazil." Journal of Transport & Health 15 (2019): 100658.

2. Maslamani, A., Almagbile, A., & Dayafleh, O. (2021). Analysis of Spatial Accessibility and Capacity of Multi-Level Healthcare Facilities in Greater Irbid Municipality, Jordan. International Journal of Geoinformatics, 17(6), 111-121.

3. Hu, Wei, Lin Li, and Mo Su. "Spatial inequity of multi-level healthcare services in a rapid expanding immigrant city of China: a case study of Shenzhen." International journal of environmental research and public health 16, no. 18 (2019): 3441.

Comment 2: It will be better to add a study area map with lat. and long., so that international reader can get locational information precisely.

Response 2: Thanks. We have included study area map in the revised manuscript.

Figure 1. Location of public hospitals with their bed capacity in Jeddah city, KSA.

Comment 3: Road features considered in this study are believed to be highly useful in developing a successful location-allocation model. Did you face any topological errors in the road network? If yes, then how you managed the error in the datasets? Please state it in your work.

Response 3: Thank you for your comment. We agree that road features are highly useful in developing a successful location-allocation model. In this study, we used the Jeddah Municipality Road Network map dataset that is similar to OpenStreetMap road network dataset. We encountered some topological errors whilst working, such as roads that were disconnected or had incorrect attributes. We managed these errors by manually inspecting the dataset and correcting the errors, wehre appropriate. In addition, we developed topology datasets in ArcGIS (v. 10.5) to get rid of certain topological defects (e.g., overextended line and gap between lines). The layers were then used to create a network dataset within a geodatabase.

Comment 4: Figures 1 and 2 unnecessarily increased the volume of the paper. I’d suggest combining them in a single figure.

Response 4: Thanks. We have combined Figs. 1 and 2 in the revised manuscript. 

Figure 1. Location of public hospitals with their bed capacity in Jeddah city, KSA.

Comment 5: Are the results (or the method) sensitive to this specific study area or can be scaled up?

Response 5: Thanks. The study has wider ramifications given urban populations are skyrocketing globallyup. Although the results of a location-allocation model are often limited to the study area and may not be readily transferable to other areas or cities, we believe the method can be scaled up to other setting. The above statements were also included in the revised manuscript. Please see the page 20 and lines 325-333.

Comment 6: Regarding the solutions, are these solutions Feasible or Optimal solutions? Please make this distinction both in the results and in their discussion.

Response 6: Thanks for your comment. We can infer that the solutions presented in the study are feasible in the sense that they result in the allocation of hospitals that effectively serve a substantial number of districts in the study area. However, they may not be considered optimal because there is a need to establish additional hospitals to cover all residents of the city. The discussion of the study would likely emphasize the feasibility of the proposed solutions in terms of improved coverage achieved by the allocated hospitals. However, it would also highlight the need for further optimization to ensure that the entire city receives timely and adequate healthcare services. The study might suggest that the current allocation, while feasible to some extent, falls short of being fully optimal in meeting the healthcare demands of the growing population. Please see the pages 20-21 and lines 334-343.

Comment 7: Please make sure your conclusions section underscores the scientific value added to your paper and/or the applicability of your findings/results. It would help if you enhanced your contributions and limitations, underscored the scientific value added to your paper, and/or the applicability of your findings/results and future study in this session.

Response 7: Thanks again. We have revised our conclusions according to your suggestion in the revised manuscript. The following texts are added (please see pages 21-22, lines 351-364). 

“The approach employed in this study has important policy implications for future health management planning, not only in Jeddah city but also in similar contexts. Its service-agnostic nature means that the findings can be applied to various healthcare services beyond hospitals, such as clinics or specialized healthcare centers. This flexibility allows policymakers to adapt this approach to different healthcare needs and optimize the spatial distribution of services accordingly. Overall, this study not only contributes to the improvement of health services distribution but also highlights the importance of interdisciplinary assessment. By bringing together professionals from various fields, policymakers can effectively address challenges of healthcare provision and ultimately enhance overall well-being of urban populations.”

Response to Reviewer 2 Comments

Comment: I would like to thank the authors for this interesting work. Please consider shortening the paper by avoiding repetition and too much theoretical descriptions that could be resolved using brief description and references to original studies introducing the theories/concepts. My detailed comments can be found below:

Response: The authors would like to thank the reviewer for his excellent suggestions and comments. We are highly indebted to your valuable comments, which improved the quality of the manuscript. 

The manuscript has been rechecked and appropriate changes have been made in accordance with the reviewers’ suggestions. A response document is prepared to indicate changes in the manuscript relative to the comments. Corrected/inserted/amended texts are highlighted in YELLOW in the revised manuscript. 

Comment A: Abstract: “better serve better 94 districts” – please check

Response A: Thanks. It was amended in the revised manuscript. Please see the page 1 and lines 22-23.

Comment B: Introduction: The introduction is needlessly long. It would be better to remove some components that explain the applied model to the methodology section. The parts that could be considered for moving are listed below:

Response B: Thank you. We have modified our introduction section as per your suggestion.

Comment 1: “This work proposed a novel method…facility or service”

Response 1: Thanks. It was amended in the revised manuscript. Please see the page 6 and lines 128.

Comment 2: “Based on the preceding discussion…facilities?” – the research question should come at the end of this section.

Response 2: We have modified structure of the introduction section to enahnce readability. Please see the revised introduction section (pages 5-6, lines 100-126). 

Comment 3: “The fuzzy overlay toolset…less-demand sites”

Additionally, please include a condensed literature review (a single para should suffice) that compares the existing LA model and how this method is novel or a new addition to the knowledge base. Moreover, the introduction does not have a meaningful flow. Please revise and reorganize.

Response 3: We have modified structure of the introduction section to accommodate your observation (please see pages 5-6, lines 100-126).

Comment C: Methods

-Consider changing the heading of 2.1 to Study Area or a similar simpler heading

Response C: We have changed the heading in the revised manuscript. 

Comment 1: 2.1 First para requires some references.

Response 1: We have included references according to your comment.

Comment 2: 2.1 A map of the study area could be useful.

Response 2: A study area map has been added.

Figure 1. Location of public hospitals with their bed capacity in Jeddah city, KSA.

Comment 3: 2.3 Consider renaming the heading. Should be ‘Data Analysis’ or ‘MCDM Process’ or words similar to that effect. This is because all your sub-sections under section 2 are part of the methods.

Response 3: Heading and sub-heading have been revised accordingly.

Comment 3: 2.3 ArcGIS, which version?

Response 4: ArcGIS (V.10.5).

Comment 5: 2.3 When explaining the algorithm that was used to determine the default kernel, the term ‘weighted’ was used. It is important to mention what this implies and what kind of weight assignments were performed.

Response 5: Thanks for your comment. The specific weight assignments performed in the context of kernel determination can vary depending on the algorithm and the problem domain. However, the weights are generally used to assign relative importance or influence to different elements or factors within the algorithm. We have considered that all the elements have the same importance, so the weight would be the same for all, i.e., 1 in this case.

Comment 6: 2.3.2 Condense. There are too many introductory descriptions here. Figure 1 could be placed in 2.1 to give general readers an idea about the study area before the variables are introduced in the subsequent parts. Jeddah’s boundary should be included in the Figure.

Response 6: We have added a study area map in the revised manuscript. 

Figure 1. Location of public hospitals with their bed capacity in Jeddah city, KSA.

Comment 7: 2.3.2 Figure 1 and 2 could be merged. Consider showing the point locations of the hospitals and color gradation based on hospital numbers, as in Figure 2. The labels could be turned on to display hospital names on the points.

Response 7: Figures 1 and 2 were combined according to your suggestion in the revised manuscript.

Figure 1. Location of public hospitals with their bed capacity in Jeddah city, KSA.

Comment 8: 2.3.4 Again, such a detailed description of the location-allocation model and its applications are unnecessary. Focus on the P-median model that you have applied, how it is novel, and how it was implemented in a GIS platform.

Response 8: We have revised section 2.3.4 according to your suggestion. Besides, we have included a flowchart of the P-Median problem workflow in the revision.

Figure 2. Schematic diagram, showing P-Median solution.

Comment 9: 2.3.4 Consider moving the mathematical description to a supplementary file. The paper is not method intensive. Therefore, too much focus on the mathematical model might distract the readers from the main objectives. Also, if you choose to include mathematical notations, there must be equation numbers, and every function (English alphabet) must be expressed with what they stand for. For example, it is unclear what the ‘a’, ‘d’, ‘x’ etc., stand for in the objective function. This description was provided at the end but every description must come underneath the equation when the function was first introduced. This part raises more questions than it answers. I would suggest simplifying this part and instead focusing on how this model was implemented by providing references to the P-median MCDM model from earlier studies as references.

Response 9: We have modified this section to enhance clarity and readability. 

Comment 10: 2.3.4 Instead of such a detailed mathematical description, consider including a schematic diagram and its description to explain the study design or P-median implementation.

Response 10: We have included a flowchart of the P-Median workflow.

Figure 2. Schematic diagram, showing P-Median solution.

Comment 11: MCDM generally has a scaling factor applied to combine the multi-criteria variables. Please mention if this was done, and if yes, how.

Response 11: Thank you. In this study, min-max scaling method was considered. This method rescales the criteria values linearly to a common range, typically between 0 and 1. The formula for min-max scaling is:

Scaled value = (value-min) / (max-min)

Here, “value” represents the original value of a criterion, “min” is the minimum value observed for that criterion, and “max” is the maximum value observed. This scaling ensures that all criteria have a similar range and avoids giving undue importance to criteria with large values.

Comment 12: Figure 3: Scalebar has problems. The numbers should be spread out more in the beginning part.

Response 12: Thanks. The problem is addressed in the revision.

Figure 3. Number of beds in hospital derived by kernel density.

Comment 13: 3.2 First para repeats the methods. No need.

Response 13: Amended.

Comment 14: 3.2 Third and fourth para could be used to replace section 2.3.4

Response 14: Done.

Comment 15: Figure 6 (e.g., Candidate) and 7 (e.g., restriction) have symbols that do not exist in the map. Please revise. What does ‘lines’ mean?

Response 15: It was amended in the revised manuscript. 

Comment 16: Conclusion Should be rewritten. The focus should not be on the application of GIS or fuzzy overlay or p-median. Rather, what you found in the study, specifically, what discrepancies in existing health facilities locations were revealed through the study analysis?

Response 16: We have revised our conclusions as per your suggestion. 

Response to Reviewer 3 Comments

The manuscript presents a novel approach for the optimization of hospital locations in urbanized regions like Jeddah City, Saudi Arabia, based on a multi-criteria decision model (MCDM) and the P-median location-allocation model. By combining these techniques into an MC-P-median optimizer, the authors aim to aid health planners in determining the best locations for public hospitals. The study was conducted using a geodatabase of public hospitals, road networks, and population districts in Jeddah, with results suggesting the need for additional hospitals for comprehensive coverage.

Response: The authors would like to thank the reviewer for his excellent suggestions and comments. We are highly indebted to your valuable comments, which improved the quality of the manuscript. 

Major Concerns:

Comment 1: The major concern with this manuscript is its structure and organization, which presently do not facilitate a clear understanding of the work's aims, objectives, or hypotheses, particularly how these are addressed in the research findings.

Specifically, the authors should consider separating the results and discussion sections to clearly present and then interpret their findings. By doing so, they can more effectively contextualize their results within the broader literature, address the implications of their study, and discuss any limitations that might impact the applicability of their research in real-world settings.

Response 1: The reviewer’s concerns regarding the manuscript’s structure and organization are taken care of. Therefore, we have addressed these issues and improved clarity and coherence of the work. However, separating the results and discussion sections would make our manuscript less readable. Hence, we added them in one section. 

Minor Observations:

Comment 2: It's unclear from the manuscript whether the road network data was used in distance calculations.

Response 2: We have apologized for this vague statement. The road network data was used in distance calculations. Basically, in this study, two types of impedance cutoffs (cost functions) were considered: travel distance (km) and travel time (minutes).

Comment 3: The figure captions need to be improved for better clarity and to guide the reader’s focus within the figure.

Response 3: They have been amended in the revised manuscript. 

Comment 4: The authors mention localities adjacent to Jeddah City in the problem setting, but the study appears limited to the city's boundaries. The impact of this exclusion on the model's outcomes should be addressed.

Response 4: You are correct in pointing out that the study appears to be limited to the boundaries of Jeddah city, despite mentioning of adjacent localities in the problem set. The authors acknowledged this limitation and discussed its implications for the model’s outcomes and generalizability in the revised manuscript. 

By excluding adjacent localities from the study area, the results may not fully capture broader healthcare needs and dynamics of the entire region. However, this limitation could not affect overall accuracy and reliability of the optimized locations for healthcare facilities, as the model may not account for potential demand from neighbouring areas or the influence of population movement across boundaries. Moreover, to address this issue, the authors emphasized the need for further research.

Comment 5: The text following the aim statement in the introduction seems redundant and could potentially be removed.

Response 5: It was amended.

Comment 6: The authors should consider merging Figures 1 and 2 to better convey the number of beds information. A consistent, rounded scale could also improve reader comprehension.

Response 6: Figures 1 and 2 were combined according to your suggestion.

Figure 1. Location of public hospitals with their bed capacity in Jeddah city, KSA.

Comment 7: On page 14, the formula for a linear programming problem should be contextualized for the current problem, explaining how various parameters in the equation are applicable in this context.

Response 7: Thanks. Considering your comment and in line with improving readability of methods section, the authors have deleted the formula and instead include a flowchart illustrating the workflow of P-Median solution. The graphical representation provides a concise methodological overview, enabling researchers to grasp the key steps. Ultimately, this revision contributes to overall readability of the manuscript and enhances clarity of the study’s approach to the readers.

Comment 8: The scale inconsistency across Figures 3 to 7 can be confusing for readers. The authors should ensure the scales remain consistent for the same aerial extent.

Response 8: We have revised all the figures in the revision and ensured the scales remain consistent for the same aerial extent. 

Comment 9: Figures 3 to 5 lack internal reference points to aid reader comparison and understanding. The inclusion of recognizable features, such as significant points or road junctions, could improve this.

Response 9: We have revised all the figures and eliminated redundant features or symbols to improve readability. 

Comment 10: In conclusion, while the manuscript offers a promising contribution to the field of health service optimization, certain adjustments are necessary to enhance the clarity and impact of the findings. It is recommended that the authors carefully address the above concerns and observations in their revisions.

Response 10: Thanks for your suggestion. According to your recommendation, we have carefully addressed your concerns and observations in the revised manuscript.

---

## [Decision Letter · Decision Letter 1]

6 Nov 2023

PONE-D-23-14949R1Optimizing health service location in a highly urbanized city: multi criteria -P-Median model for public hospitals in Jeddah city, KSAPLOS ONE

Dear Dr. Murad,

Thank you for submitting your manuscript to PLOS ONE. After careful consideration, we feel that it has merit but does not fully meet PLOS ONE’s publication criteria as it currently stands. Therefore, we invite you to submit a revised version of the manuscript that addresses the points raised during the review process.

We look forward to receiving your revised manuscript.

Kind regards,

Mohammed Sarfaraz Gani Adnan, PhD

Academic Editor

PLOS ONE

Journal Requirements:

Additional Editor Comments:Given the combination of the results and discussion sections, it is essential to provide comprehensive and critical discussions of the results obtained in this study. Such discussions are necessary to validate and justify the key findings of this research.Regarding Figure 3-5, it is recommended to ensure consistency in the layout, such as positioning the mapping components in a similar fashion. Additionally, consider the possibility of combining these figures into one layout for clarity. Overlaying the locations of hospitals on these maps would enhance the reference points for readers, particularly when making geographical comparisons with Figure 1.For the classification range, it is advisable to reconsider the current classes, which appear somewhat arbitrary (e.g., 6781.93 to 9040.88). A more reader-friendly and intuitive approach would be to use rounded ranges, such as 6500 to 9000, which will be easier for readers to comprehend.

Reviewers' comments:

Reviewer's Responses to Questions

**Comments to the Author**

1. If the authors have adequately addressed your comments raised in a previous round of review and you feel that this manuscript is now acceptable for publication, you may indicate that here to bypass the “Comments to the Author” section, enter your conflict of interest statement in the “Confidential to Editor” section, and submit your "Accept" recommendation.

Reviewer #2: All comments have been addressed

Reviewer #3: All comments have been addressed

2. Is the manuscript technically sound, and do the data support the conclusions?

Reviewer #2: Yes

Reviewer #3: Yes

3. Has the statistical analysis been performed appropriately and rigorously? 

Reviewer #2: Yes

Reviewer #3: Yes

4. Have the authors made all data underlying the findings in their manuscript fully available?

Reviewer #2: Yes

Reviewer #3: Yes

5. Is the manuscript presented in an intelligible fashion and written in standard English?

Reviewer #2: Yes

Reviewer #3: Yes

6. Review Comments to the Author

Reviewer #2: Thanks for addressing the comments. It's in much better shape now.

Please check the journal guidelines for further instructions on scope of improvement.

Reviewer #3: (No Response)

7. PLOS authors have the option to publish the peer review history of their article (what does this mean?). If published, this will include your full peer review and any attached files.

Reviewer #2: No

Reviewer #3: No

---

## [Author Response · Author response to Decision Letter 1]

8 Nov 2023

Response to the editor comments

Dear Academic Editor

I am pleased to resubmit to you my new paper titled: Optimizing health service location in a highly urbanized city: multi criteria -P-Median model for public hospitals in Jeddah city, KSA

We have carefully reviewed the editor’s comments and addressed almost them in the revised manuscript. All revised portions are marked in YELLOW in the revised manuscript, which we would like to submit for your kind consideration. The authors pay special thanks to the editor and reviewer because their comments and suggestions have greatly improved quality of the manuscript. 

I hope that this paper can be accepted for your journal.

Best Regards

Prof. Abdulkader Murad

Additional Editor Comments:

Comment 1: Given the combination of the results and discussion sections, it is essential to provide comprehensive and critical discussions of the results obtained in this study. Such discussions are necessary to validate and justify the key findings of this research.

Response 1: Thank you for your comments. The comprehensive discussion of the study’s results validates the critical importance of addressing healthcare access disparities in urban areas, as revealed by the spatial distribution of resources in Jeddah. The selection of the fuzzy ‘And’ model, P-Median model, and the integration of the MC-P-Median model with fuzzy overlay is justified by their ability to optimize resource allocation, tackle uncertainty, and eliminate redundancy. The findings emphasize that while the current allocation is deemed feasible, further optimization is imperative to meet the healthcare demands of Jeddah’s growing population. These results have broader implications for urban healthcare planning globally, underscoring the need for data-driven, context-specific strategies and advanced spatial decision support systems to ensure equitable and efficient healthcare services as urban populations continue to expand. Please see pages 20-21 and lines 354-381.

…. Added discussion….

“One of the key takeaways from this study is the utilization of a GIS for healthcare planning. Integration of GIS technology with hospital locations enabled a detailed analysis of the distribution of healthcare resources and service demands. Combining P-Median model and fuzzy overlay techniques in a spatial support system addressed complex issues related to healthcare access and resource allocation. Results emphasized the importance of population density, the proximity of healthcare facilities, and travel times in healthcare planning. This approach recognized that healthcare accessibility is not solely dependent on number of hospitals but also on their strategic placement relative to population centers. Furthermore, the study also underscored significance of understanding local context, including factors like available resources, population density, and nature of demand locations when determining travel time thresholds. This adaptability and context-based decision-making are vital for effective healthcare planning.

The study asserts that integration of fuzzy overlay with MC-P-Median approach is superior to traditional methods, such as buffer analysis or distance-based catchment modelling. The findings related to the need for additional hospitals to cover all residents in Jeddah are a crucial finding and can help policy makers to enhance healthcare facilities situation. While the current allocation is deemed feasible, it is acknowledged that further optimization is necessary to ensure comprehensive healthcare coverage for the growing population. This raises questions about the scalability of the proposed models to other urban areas with increasing populations, emphasizing the need for ongoing research and adaptability in healthcare planning strategies.

This work has some limitations including its focus was solely on Jeddah city. This limitation may affect comprehensive understanding of healthcare needs and dynamics in the broader region. The discussion should have the implications for not accounting for potential demand from neighbouring areas and the importance of regional dynamics in healthcare planning. In conclusion, this study offers a comprehensive and innovative approach to allocating healthcare facility and optimizing in urban areas. However, the findings underscored the importance of data-driven and context-specific strategies in addressing complex issue associated with healthcare accessibility for growing urban populations.”

Comment 2: Regarding Figure 3-5, it is recommended to ensure consistency in the layout, such as positioning the mapping components in a similar fashion. Additionally, consider the possibility of combining these figures into one layout for clarity. Overlaying the locations of hospitals on these maps would enhance the reference points for readers, particularly when making geographical comparisons with Figure 1.

Response 2: We greatly appreciate the editor’s valuable input regarding the layout and presentation of our figures. We have agreed that consistency in the positioning of mapping components is essential for a coherent presentation. We have ensured that the layout of Figure 3-5 has been standardized to make it easier for readers to navigate and compare the information. Additionally, we have explored the possibility of combining these figures into one layout to enhance clarity and facilitate geographical comparisons, as suggested. We have also overlaid hospital locations on these maps to provide readers with clear reference points, which has improved the accessibility of our findings.

Figure 3. a. Number of beds in hospital derived by Kernel density. b. Population density in Jeddah as a measure of health demand. c. A provider-to-population result-based fuzzy overlay analysis. The public hospitals better serve locations in red, while locations in green need additional hospitals (The figure made with ArcGIS v.10.5 software (https://www.esri.com/en-us/arcgis/products/arcgis-desktop/resources)). [Shape file or base map that is freely available at https://www.diva-gis.org/gdata and do not have copyright restrictions.]

Comment 3: For the classification range, it is advisable to reconsider the current classes, which appear somewhat arbitrary (e.g., 6781.93 to 9040.88). A more reader-friendly and intuitive approach would be to use rounded ranges, such as 6500 to 9000, which will be easier for readers to comprehend.

Response 3: We appreciate the editor’s feedback regarding the classification range in our study. We agree that using more reader-friendly and rounded ranges, such as 6500 to 9000, could enhance the clarity and comprehension of our data. We have revisited the classification ranges and made the necessary adjustments to ensure that our readers can easily interpret the information in the revised manuscript.

Figure 3. a. Number of beds in hospital derived by Kernel density. b. Population density in Jeddah as a measure of health demand. c. A provider-to-population result-based fuzzy overlay analysis. The public hospitals better serve locations in red, while locations in green need additional hospitals (The figure made with ArcGIS v.10.5 software (https://www.esri.com/en-us/arcgis/products/arcgis-desktop/resources)). [Shape file or base map that is freely available at https://www.diva-gis.org/gdata and do not have copyright restrictions.]

---

## [Editor Report · Decision Letter 2]

10 Nov 2023

Optimizing health service location in a highly urbanized city: multi criteria -P-Median model for public hospitals in Jeddah city, KSA

PONE-D-23-14949R2

Dear Dr. Murad,

We’re pleased to inform you that your manuscript has been judged scientifically suitable for publication and will be formally accepted for publication once it meets all outstanding technical requirements.

Kind regards,

Mohammed Sarfaraz Gani Adnan, PhD

Academic Editor

PLOS ONE
---

## [Editor Report · Acceptance letter]

20 Dec 2023

PONE-D-23-14949R2 

PLOS ONE

Dear Dr. Murad, 

I'm pleased to inform you that your manuscript has been deemed suitable for publication in PLOS ONE. Congratulations! Your manuscript is now being handed over to our production team.

Kind regards, 

on behalf of

Dr. Mohammed Sarfaraz Gani Adnan 

Academic Editor

PLOS ONE